# Structural basis for the adsorption of a single-stranded RNA bacteriophage

Ran Meng [1], Mengqiu Jiang[1], Zhicheng Cui[1], Jeng-Yih Chang[1], Kailu Yang[1,4], Joanita Jakana[2], Xinzhe Yu[2], Zhao Wang [2], Bo Hu[3] & Junjie Zhang[1]

Single-stranded RNA bacteriophages (ssRNA phages) infect Gram-negative bacteria via a single maturation protein (Mat), which attaches to a retractile pilus of the host. Here we present structures of the ssRNA phage MS2 in complex with the *Escherichia coli* F-pilus, showing a network of hydrophobic and electrostatic interactions at the Mat-pilus interface. Moreover, binding of the pilus induces slight orientational variations of the Mat relative to the rest of the phage capsid, priming the Mat-connected genomic RNA (gRNA) for its release from the virions. The exposed tip of the attached Mat points opposite to the direction of the pilus retraction, which may facilitate the translocation of the gRNA from the capsid into the host cytosol. In addition, our structures determine the orientation of the assembled F-pilin subunits relative to the cell envelope, providing insights into the F-like type IV secretion systems.

[1] Department of Biochemistry and Biophysics, Center for Phage Technology, Texas A&M University, College Station, TX 77843, USA. [2] National Center for Macromolecular Imaging, Verna and Marrs McLean Department of Biochemistry and Molecular Biology, Baylor College of Medicine, Houston, TX 77030, USA. [3] Department of Microbiology and Molecular Genetics, The University of Texas Health Science Center at Houston, Houston, TX 77030, USA. [4] Present address: Howard Hughes Medical Institute, Department of Molecular and Cellular Physiology, Stanford University, Stanford, CA 94305, USA. Correspondence and requests for materials should be addressed to B.H. (email: Bo.Hu@uth.tmc.edu) or to J.Z. (email: junjiez@tamu.edu)

Single-stranded RNA bacteriophages (ssRNA phages) are positive-sense RNA viruses that infect a variety of Gram-negative bacteria, including *Escherichia coli*, *Acinetobacter spp.*, *Caulobacter crescentus*, and *Pseudomonas aeruginosa*, etc.[1–5]. The infection cycle consists of the following steps: first, the ssRNA phage adsorbs to the side of a retractile pilus via its maturation protein (Mat), presented as a single copy in the virion shell. Then the Mat, along with the genomic RNA (gRNA), is released from the viral capsid and taken up by the host, presumably through the retractile force of the host pilus[6]. Cytosolic internalization of the gRNA initiates the expression of the viral genes, resulting in replication, virion morphogenesis, and lysis of the host[7].

All known varieties of retractile pili in Proteobacteria can serve as the receptor of ssRNA phages, including conjugative pili (e.g. F and M)[6,8,9], type IV twitching motility pili[10], and species-specific pili, such as the Tad pili of swarmer *Caulobacter* cells[11]. It should be noted that ssRNA phages specific for the pili encoded on conjugal plasmids may not have a required bacterial host as these plasmids can have a wide host range[12].

The binding affinity of a particular phage to pili of the same family can vary significantly. For example, MS2 binds tightly to the F-pili encoded by the pOX38 plasmid but weakly to the pili encoded by pED208, which belongs to the same F-like plasmid family[13]. Mutations on the F-pilin can abolish the binding of the ssRNA phages, leading to phage-resistant mutants[13,14]. The underlying structural cause for the differences in adsorption of ssRNA phages is unknown, primarily due to the lack of high-resolution structures of ssRNA phages bound to their receptor pili. The only structure of phage bound to host pili is a 39 Å-resolution cryo-electron tomography (cryo-ET) subtomogram average of MS2 in complex with the purified F-pilus, in which the interactions between amino acid residues are not clear[15]. A better understanding of the adsorption requirements of ssRNA phages to their respective hosts can facilitate the engineering of new ssRNA phages to deliver foreign RNA molecules to a variety of cells[16,17].

Here, we investigate the canonical model system of MS2 and its receptor, the F-pilus, in high resolution. The capsid of MS2 is composed of a single copy of Mat and 178 coat proteins (89 coat protein dimers) with a T = 3 quasi-icosahedral symmetry[1,18,19]. The encapsidated gRNA (3569 bp) interact with the Mat and coat proteins in a defined three-dimensional (3D) conformation[1,18,19] with its 3′ end attached to the Mat[1]. The F-pilus, the receptor for MS2, is a dynamic filament assembled by the F plasmid-encoded type IV secretion system (T4SS), which directs the conjugative transfer of F plasmids[20]. The retraction of the F-pilus brings donor and recipient cells into direct contact as a prerequisite for the formation of the mating junction, through which the plasmid DNA can be transferred[21]. ssRNA phages bind to the outside of the F-pilus and apparently have no access to its inner channel[6,22]. How the gRNA is translocated across the two membranes of Gram-negative bacteria to enter the cytosol remains a mystery.

A recent high-resolution cryo-EM structure revealed that the conjugative F-pilus is an assembly of a stoichiometric pilin-phospholipid complex forming a five-start helical filament[23]. Each pilin subunit has three helices termed: α1, α2, and α3, respectively, from the N- to the C-termini (Supplementary Fig. 1a). The loop connecting α2 and α3 is located in the inner lumen of the pilus and interacts with a phospholipid, while the N- and C-termini of the pilin face outside, potentially interacting with ssRNA phages. Of note, the overall orientation of the F-pilus relative to the cell membrane is still unknown: it is unclear whether the membrane-proximal end of the F-pilus consists of the α2–α3 loops or the N-/C-termini of the pilins. This information is important to decipher the mechanism of infection for

ssRNA phages, as well as to understand the assembly/retraction dynamics of F-pili.

In this study, we present the single-particle cryo-EM structures of different conformations of the MS2/F-pilus complex at resolutions ranging from 5.6 to 7.3 Å and an average reconstruction with all particles at 5.0 Å resolution. We reveal a network of hydrophobic and electrostatic interactions at the phage–pilus interface. We also observe slight variations of the orientations for the Mat within the capsid upon pilus binding, which may prime the Mat-gRNA complex for its release. Combined with cryo-ET and subtomogram averaging of the MS2 particles that are bound to the F-pili emanating from the *E. coli* cells, our structures allow us to unambiguously determine the orientation of MS2 relative to the *E. coli* cell envelope, as well as the orientation of the F-pilin monomers in the F-pilus relative to the bacterial cell envelope.

## Results

**The overall architecture of the MS2/F-pilus complex.** One major challenge when using single-particle cryo-EM to study the MS2/F-pilus interaction is that the F-pilus is a long filamentous oligomer of pilin monomers, lying parallel to the cryo-EM grid, which leads to preferred orientations of the adsorbed phages (Fig. 1a). To minimize the range of missing orientations for the

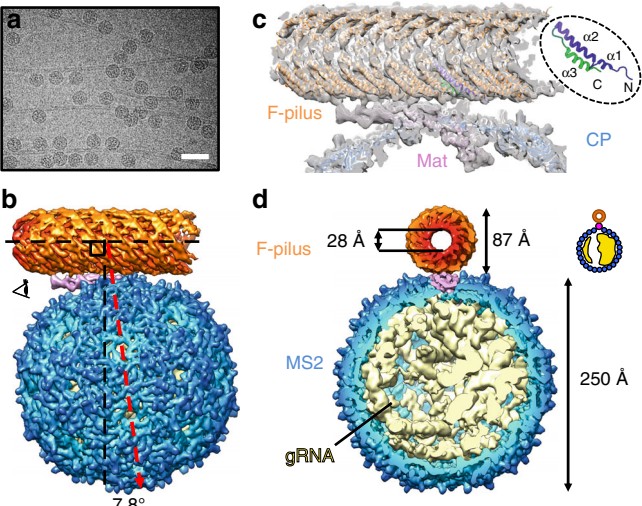

**Fig. 1** The overall architecture of the MS2/F-pilus complex. **a** An electron micrograph of the MS2/F-pilus complex. The scale bar denotes 500 Å. **b** The side view of the MS2/F-pilus complex with the F-pilus colored in orange, the Mat colored in pink, coat proteins in blue, and gRNA in light yellow. For visualization, the density map in Fig. 1 is a composite map with the density of the F-pilus/Mat complex replaced by the 6.2 Å-resolution map, which has a more uniformly well-resolved F-pilus fragment. The axis of the F-pilus is labeled by a horizontal dashed black line with another vertical dashed black line perpendicular to it. A dashed red arrow denotes the two-fold axis on the side of the MS2 capsid. The angle between the vertical dashed black line and the dashed red arrow is 7.8°. **c** The model of the Mat/F-pilus complex fit in the cryo-EM map. One of the F-pilin is colored from purple (N terminus) to green (C terminus) as in the inset to show the orientation of the F-pilins in the assembled pilus. The three α-helices of an F-pilin are also labeled in the inset. **d** Cut-in view of the density map of the MS2/F-pilus complex viewed from the direction of the eye cartoon in panel **b**. The outer and inner diameters of the F-pilus are 87 and 28 Å, respectively. The diameter of the MS2 capsid is ~250 Å. The density of the gRNA is from the refinement of all particles without any mask and low-pass filtered to 8 Å. The inset shows a cartoon model for the asymmetric distribution of the gRNA

MS2/F-pilus complex, each micrograph was collected with the specimen grid at a tilting angle of either 0°, 15°, or 30° and processed in combination (see Methods and Supplementary Fig. 2)[24]. This approach has recently been used to mitigate the preferred-orientation problem for single-particle cryo-EM[25,26]. Finally, we obtained a density map of the MS2/F-pilus complex at an overall resolution of 5.0 Å (Supplementary Fig. 3b, Supplementary Movie 1). The cryo-EM density for the two ends of the F-pilus is weak and has a lower local resolution (Supplementary Fig. 3a) compared to other parts of the map, suggesting that there is a relative motion of the F-pilus with regard to the MS2 phage particle. To improve the resolution of the F-pilus, we created a loose mask around the F-pilus and the Mat to locally align the density within this mask. This allowed us to achieve a more uniformly resolved density of the F-pilus/Mat complex at an overall resolution of 6.2 Å (Supplementary Fig. 3b). The α-helices in each pilin monomer are clearly resolved, as is the internally buried phospholipid, which interacts with the α2–α3 loop of each F-pilin (Supplementary Fig. 4a). These features are consistent with the previous structures of the F-pilus without phages bound[23].

Overall, the structure of the MS2 particle, within the MS2/F-pilus complex, is similar to what has been previously reported[1,18,19]. The particle diameter is ~250 Å and the Mat replaces one coat protein dimer at the two-fold axis of the capsid, disrupting the otherwise perfect icosahedral symmetry of the viral capsid. The outer and inner diameters of the F-pilus are 87 and 28 Å, respectively, consistent with the previous structure of the apo-state F-pilus[23]. The binding of MS2 does not break the F-pilus filament as the structure of the F-pilus is still intact (Fig. 1). We observe that the exposed surface of the β-sheet-rich region (or β-region) in the Mat lies parallel to the axis of the F-pilus and directly interacts with the N- and C-termini of the pilin subunits within the F-pilus (Fig. 1b, c) in a defined manner, in which the tip of the β-region (Supplementary Fig. 4b) leans opposite to the N- and C-termini of the F-pilins (Fig. 1c).

The gRNA inside the capsid of the F-pilus-bound MS2 is in the same conformation as previously reported for the free virions[1,18,19]. This is consistent with the fact that binding to the F-pilus alone does not liberate the gRNA[27]. The two-fold axis on the side of the MS2 capsid (the dashed red line in Fig. 1b) is not perpendicular to the axis of the pilus. Instead, it tilts 7.8° away from the tip of the β-region (Fig. 1b). Notably, when viewed from the tip of the β-region, the gRNA is more compact on the right hemisphere within the capsid compared to the left (Fig. 1d). Such an asymmetry in the gRNA packing is formed as the gRNA folds and the capsid proteins assemble around it. These asymmetric features within the single-particle cryo-EM structure stay apparent even after we low-pass filtered the map to 80 Å-resolution (Supplementary Fig. 5), so that they can serve as structural markers to distinguish the orientation of the Mat and the F-pilus in the cryo-electron tomography study of MS2 bound to the native F-pilus protruding from the cell (see below).

**Orientational variations of the pilus-bound Mat.** The lower resolution at the two distal ends of the MS2-bound F-pilus suggests structural flexibility within the MS2/F-pilus complex. To further investigate this motion, we performed multibody refinement on the MS2/F-pilus complex and principal component analysis of the variations of the conformations (see Methods)[28]. The major structural flexibility between the F-pilus and the phage particle can be described by two principal motion vectors (the first two eigenvectors), which describe the tilting and swiveling of the F-pilus relative to the phage particle (Supplementary Fig. 6a, Supplementary Movies 2 and 3).

To measure the tilting and swiveling angles of the F-pilus relative to the MS2, we performed the 3D classification of the MS2/F-pilus complex and identified three major conformations, referred to as Classes 1–3, reconstructed from 34.38%, 33.29%, and 11.99% of all the particles, respectively (see Methods)[29]. Consistent with the results from the multibody refinement and principal component analysis, these three conformations represent the relative swiveling and tilting of the F-pilus against the phage particles (Fig. 2). Class 2 represents a conformation that is similar to the average structure obtained by combining all the particles, with the two-fold axis on the side of the phage particle deviating 7.8° from the line perpendicular to the axis of the pilus (Fig. 2e). Class 1 and Class 2 have almost the same swiveling angle for the F-pilus (Fig. 2a, d). However, the F-pilus in Class 3 swivels ~4° counter-clockwise relative to Classes 1 and 2 when viewed from the F-pilus-side of the complex (Fig. 2g). Compared with Class 2, Classes 1 and 3 represent two conformations with the pilus tilting ~2° towards or away from the tip of the β-region, respectively (Fig. 2b, h). The tilting of the F-pilus in Class 3 is so strong that it disrupts the connection between the pilus and the tip of the β-region (the black arrow in Fig. 2i and Supplementary Fig. 6b), while this connection is preserved in Classes 1 and 2.

The relative motions of the F-pilus against the MS2 virion result in a slight difference in the positioning of the attached Mat within the MS2 capsid. The values of the angle between the β-region of the Mat and the phage capsid range from 28.5° to 32.3° (Fig. 2c, f, i, Supplementary Fig. 6b). Of note, there is relatively little internal plasticity within the Mat, and at our current resolutions of the maps, there is no detectable variation in the tertiary structure of the Mat in these three classes of conformations (Supplementary Fig. 7). Such a variation for the positioning of the Mat is not observed in our control experiment, imaging the pilus-free MS2, with most of the particles (93%) in the apo MS2 have the Mat in a position angle the same as in Class 2 (Supplementary Fig. 6b). The other 7% of the particles either do not show the Mat (6%) or are of too small a number (1%) to generate good density maps.

**Interactions at the pilus–Mat interface.** The interaction between the MS2 and the F-pilus is strong, to an extent that most of the Mat proteins tend to remain associated with the F-pili even after a severe sonication[30]. The area of the interface between the β-region of the Mat and the pilus is ~1062.89 Å². Within the interface between the F-pilus and MS2, four points of interaction on the surface of four pilin subunits are identified (labeled by a black sphere and three aspartic acid residues, respectively in Fig. 3a). At the pilus–Mat interface, residues of opposite charges have electrostatic interactions. The positively charged His357 from the Mat interacts with the Asp7 from one pilin (Fig. 3b). Two arginine residues, Arg36 and Arg99 from the Mat interact with two aspartic acids, Asp7 and Asp23, from the pilins, respectively (Fig. 3c). It is interesting to note that a D23G (Asp to Gly) mutant pilin still assembles F-pili; but the assembled pili do not bind to phage R17[13]. The protein sequences of the Mat for R17 and MS2 are 97% identical and have these positively charged residues conserved (Supplementary Fig. 1b). It is likely the mutation from an aspartic acid to a glycine has compromised the electrostatic interactions between the F-pilus and the Mat. Another pilin mutant, A18E (Ala to Glu), similarly assembles R17-resistant F-pili[13]. Since Ala18 is within the pilus–Mat interface, the change to a larger sidechain of the glutamic acid likely interferes with the Mat binding (Supplementary Fig. 4b). The first five amino acids (AGSSG, Ala-Gly-Ser-Ser-Gly) at the N terminus of the mature F-pilin are generally flexible in the apo F-pilus and could not be resolved in previous work[23]. In contrast, in

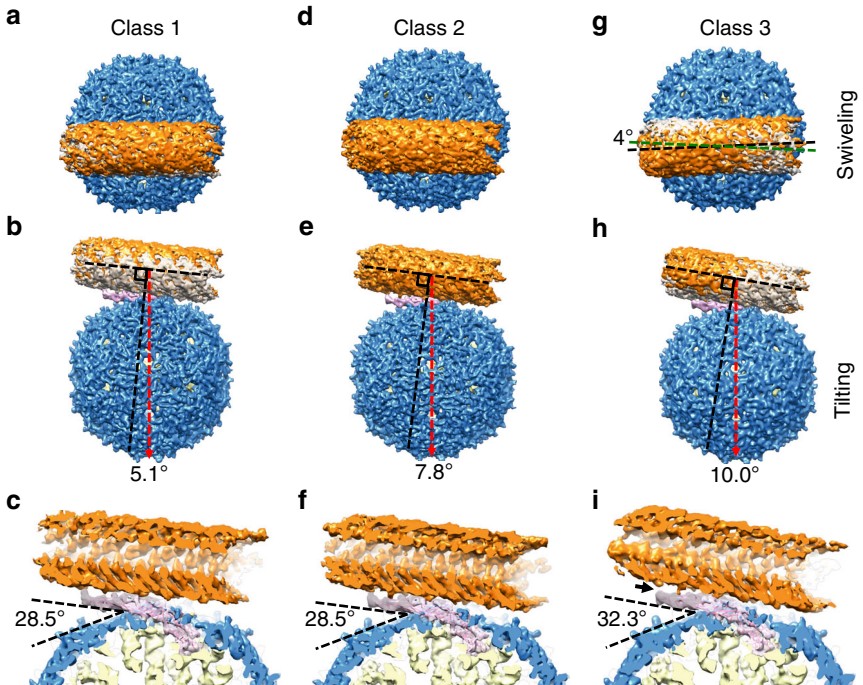

**Fig. 2** Three conformations of the MS2/F-pilus complex. **a**, **d**, **g** Top views of the MS2/F-pilus complex for Classes 1, 2, and 3. Class 1 and Class 3 are superimposed with Class 2 (gray density) for comparison in panels **a** and **g**. The dashed lines in green and black in panel **g** represent the axes of the F-pilus in Class 2 and 3. **b**, **e**, **h** Side views of the MS2/F-pilus complex for Classes 1, 2, and 3. Class 1 and Class 3 are superimposed with Class 2 (gray density) for comparison in panels **b** and **h**. The axis of the F-pilus is labeled by a dashed black line (parallel with the axis of the F-pilus) with another dashed black line perpendicular to it. The dashed red arrow denotes the two-fold axis on the side of the capsid. The angles between the perpendicular black line and the dashed red arrow are 5.1°, 7.8°, and 10.0° for Classes 1, 2, and 3, respectively. **c**, **f**, **i** Cut-in views from the side for Classes 1, 2, and 3. The angles between the tip of the β-region and phage capsid are 28.5°, 28.5°, and 32.3° in Classes 1, 2, and 3, respectively. A solid black arrow in panel **i** indicates the detachment of the tip of the β-region from the F-pilus

the phage-bound structure, we see extra cryo-EM density extending from the N terminus of one pilin subunit into a hydrophobic pocket (consisting of Val15, Phe31, Leu33, Phe92, and Phe94) on the exposed surface of the Mat (Fig. 3d, Supplementary Fig. 4c). This electron density may correspond to the five N-terminal amino acids of a pilin subunit, which were previously missing due to their flexibility but are now stabilized by the Mat. These five amino acids are present in F-pilins produced by the pOX38 plasmid but absent in F-pilins produced by pED208 (ref. [31]) (Supplementary Fig. 1a), which may contribute to the weakened binding of MS2 to pED208-encoded pili[13]. Interestingly, a loop at the tip of the β-region connects to the N terminus of an adjacent F-pilin subunit. Such a connection is present in the conformations of Classes 1 and 2 but disrupted in the conformation described in Class 3 (Supplementary Fig. 6b), due to an over-tilting of the F-pilus away from the tip of the β-region. The above-mentioned hydrophobic and positively charged amino acids in Mats, mediating the F-pilus binding, are conserved in four out of the five MS2-like phages, including MS2, R17, M12, and JP501. Fr, whose Mat sequence is less homogeneous to MS2, has the Phe92 and Phe94 missing, but still has the Val15, Leu31, and Leu33, partially preserving the hydrophobic pocket. The positively charged amino acids are conserved in Fr (Supplementary Fig. 1b).

**The bound Mat orients opposite to the pilus retraction.** In our single-particle cryo-EM structure of the complex formed by MS2 with purified F-pili, we have determined that each MS2 particle binds to the F-pilus in a defined orientation with the tip of the β-region pointing opposite to the N- and C-termini of the F-pilins.

However, the orientation of the F-pilus relative to the cell envelope is still a mystery[23]. Specifically, it is unknown if pilins are oriented so that the α2–α3 loops, or the two termini of the pilins, are closer to the cell envelope. This orientation has important implications regarding the mechanism of the F-pilus assembly and retraction. Definition of this orientation would also establish whether, during pilus retraction, the adsorbed phage particle approaches the cell with the tip of its Mat facing the cell or the opposite. To address this issue, we used cryo-ET to image MS2 adsorbed to pOX38 F-pili extruded from *E. coli* cells. The reconstructed tomograms showed that the phage particles are tightly adsorbed to the F-pilus emanating from the cell envelope (Fig. 4a). We are able to determine an averaged cryo-ET structure of the MS2/F-pilus complex in situ at 32.3 Å resolution (Fig. 4c, d, Supplementary Fig. 3b).

To analyze the relative orientation of MS2 particles to the cell envelope, we mapped the subtomogram averages back to the original tomograms (Fig. 4b). The relatively small size of the Mat may lead to ambiguity in determining its orientation relative to the cell at the current resolution of our cryo-ET subtomogram average. Therefore, we relied on features that are still apparent, even at low resolution, to verify the orientation of the phage particle against the cell. These include the asymmetric features in the MS2/F-pilus complex such as the tilting angle of the particle relative to the pilus and the uneven distribution of the gRNA inside the capsid (Supplementary Fig. 5) as shown in our single-particle structure of the MS2/F-pilus complex. Since the subtomogram average structure is solved by binding MS2 to the cell-connected F-pilus, the cell proximal end of the F-pilus is known to be on the right in Fig. 5a. With the β-region of the Mat attached parallel to the axis of the pilus, the MS2 capsid leans

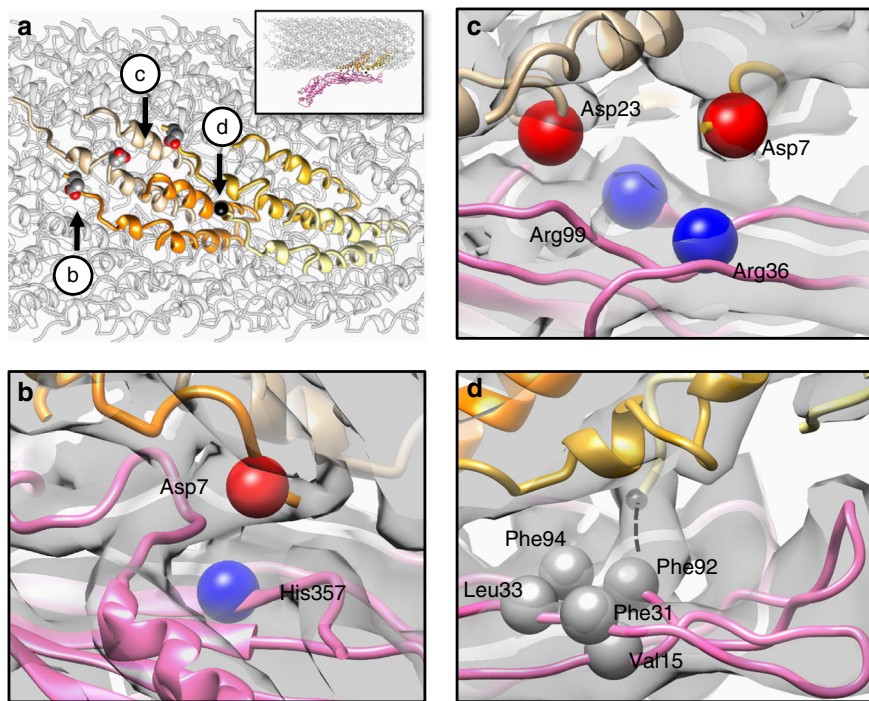

**Fig. 3** The molecular interface between MS2 and the F-pilus. **a** Four F-pilin subunits, which have specific interactions with the Mat, are colored tan, orange, gold, and khaki, respectively. The inset shows the overall side view of the F-pilus/Mat complex and the location of these four F-pilins within the pilus. Three aspartic acids are labeled as sphere models and the N terminus of one pilin is labeled by a black sphere. Arrows indicate the viewing directions for panels **b–d**. **b** The negatively charged residue Asp7 (red bead) from one F-pilin is in close proximity to the positively charged His357 (blue bead) from the Mat. **c** The negatively charged residues Asp7 and Asp23 (red beads) from two F-pilins are in close proximity to Arg36 and Arg99 (blue beads) from the Mat, respectively. **d** The N terminus of one pilin extends into a hydrophobic pocket consisting of five hydrophobic residues (Val15, Phe31, Leu33, Phe92, and Phe94, labeled by gray beads) from the Mat. The dashed black line denotes the location of the five missing residues (AGSSG) at the N terminus of one pilin. The cryo-EM density map is shown as a gray transparent envelope

towards the cell envelope, leaving the two-fold symmetry axis (the dashed red line) tilting to the opposite side of the tip of the β-region. This is consistent with the single-particle cryo-EM structure of the Mat/F-pilus complex, in which the phage capsid leans opposite to the tip of the β-region (Fig. 5b). In addition, when viewed against the tip of the β-region, the encapsidated gRNA (the yellow density in Fig. 5d) is denser on the right side of the capsid. Such a pattern of the unevenly distributed gRNA is also observed in the subtomogram average of the MS2/F-pilus complex (Fig. 5c), further confirming the orientation of the Mat within the tomogram. We conclude that when MS2 is adsorbed to the side of the F-pilus, the tip of its Mat is pointed toward the tip of the F-pilus and opposite the cell envelope. These findings also allow us to orient the pilins within the F-pilus relative to the cell envelope: the N- and C-termini of the pilins are positioned more closely than the phospholipid/central loop to the cell envelope (Fig. 6).

## Discussion

In this paper, we have revealed the molecular and structural requirements for adsorption of the phage MS2 onto the wild-type F-pili. A network of electrostatic and hydrophobic interactions is identified between the conjugative F-pilus and the Mat of MS2. As the conjugative plasmids for the pili are subject to strong selection for mutations that confer phage resistance yet allow conjugation, and the ssRNA phages evolve to suppress the resistance, such a pattern of interacting protein residues (Supplementary Fig. 1) may have evolved to maintain a balance between the phage and its host. Given the genetic power of the *E. coli*/ssRNA phage system, the structural data from this work can provide more information on the phage-receptor interaction than

is available for any other prokaryotic viral system, and may provide a unique platform for probing the evolution of virus–host interactions.

As shown in our cryo-electron tomograms of the MS2 infecting the F$^+$ *E. coli* cell, most of the phages are tightly bound to the F-pilus. F-pili naturally extend and retract through the polymerization and depolymerization of the F-pilin subunits in the T4SS basal body, processes that involve rotation of the pilus[6]. During ssRNA phage adsorption, the Mat binds to the side of the pilus through interactions involving hydrophobic and charged amino acids that are strong enough to maintain the attachment of the phage particles during retraction and rotation. It was reported that when ssRNA phage R17 infects the cell, its Mat (39 kDa) is cleaved into two fragments (24 and 15 kDa, respectively). This cleavage reaction is presumed to occur after binding to the F-pilus, but before the Mat and the gRNA enter the cell[30]. Such a processing reaction may facilitate Mat's ability to cross the cell envelope. The orientation we defined here for pilus-bound Mat may provide a cell-associate protease access to a cleavage site in the Mat during pilus retraction[32]. Such an orientation of the Mat, with its tip opposite to the direction of the pilus retraction (Fig. 6a), may also facilitate detachment of the Mat from the capsid, as if it were a pop tab on a soda can (Fig. 6c). Finally, we observed that the binding of the F-pilus causes the variations in the orientations for the Mat relative to the neighboring coat proteins (Fig. 6b). As the Mat is still attached to the gRNA, the flexibility induced by the pilus binding may prime the Mat-gRNA complex to leave the rest of the capsid.

For most of MS2 phage particles bound to either purified or cell-connected F-pili, gRNA remains inside the capsid. This strongly indicates that phage binding to the F-pilus does not

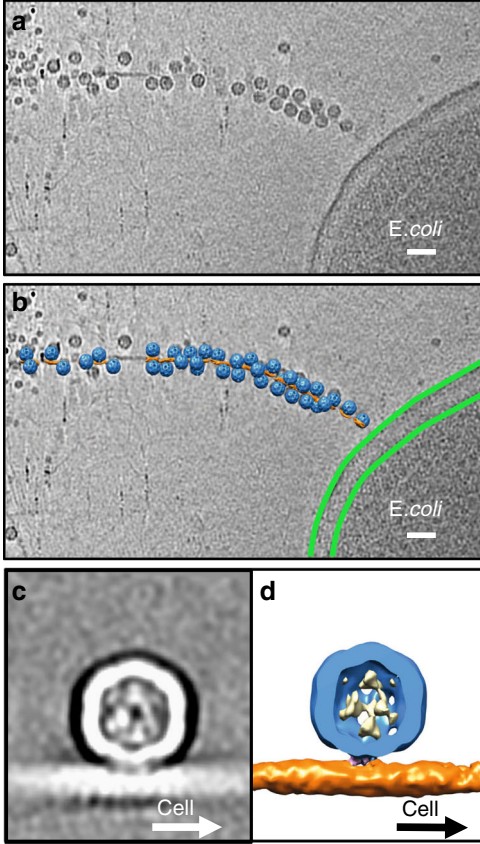

**Fig. 4** MS2 adsorbed to F-pili in situ. **a** A slice view of a tomogram. The scale bar denotes 500 Å. **b** A slice view of the tomogram superimposed with the mapped-back subtomogram averages. MS2 are colored blue. The F-pilus is colored orange. The outer and inner membranes of the *E. coli* are labeled by two green curves. The scale bar denotes 500 Å. **c** A slice view of the subtomogram average of the MS2/F-pilus complex. **d** A cut-in view of the subtomogram average with the gRNA colored yellow, coat proteins blue, the Mat pink, and the F-pilus orange

release the gRNA. Rather, as indicated above, this release event is envisioned to occur when, as a consequence of pilus retraction, the phage collides with the cell surface or the T4SS basal structure. At this point of contact, the capsid of coat proteins is blocked, whereas the released Mat-gRNA complex is able to transit the cell envelope through a yet unidentified route. Conceivably, the T4SS basal structure may provide such a routing pathway, but this remains to be tested. Previous studies established that the phage gRNA become RNase-sensitive at a specific point in the infection process[22]. Presumably, when MS2 collides with the cell envelope during pilus retraction, the released Mat-gRNA complex is transiently accessible to RNase prior to entering the cell.

In this study, we used single-particle cryo-EM at tilted angles to alleviate the preferred-orientation of the phage particles as they attach to the F-pili that lack the end-on views. Such a strategy may be used to study other macromolecules as they interact with filamentous specimens, e.g., to study motor proteins decorated on cytoskeletons[33]. The use of large asymmetrical phage particles that have firm binding to the pili may also serve as structural markers to provide an alternative strategy to determine the unknown polarity of other pili from the cells.

## Methods

**F-pili and MS2 purifications.** *E. coli* F+ strain MC4100 containing the pOX38 (Kanamycin) plasmid was used for F-pili production. The pilin gene on the pOX38

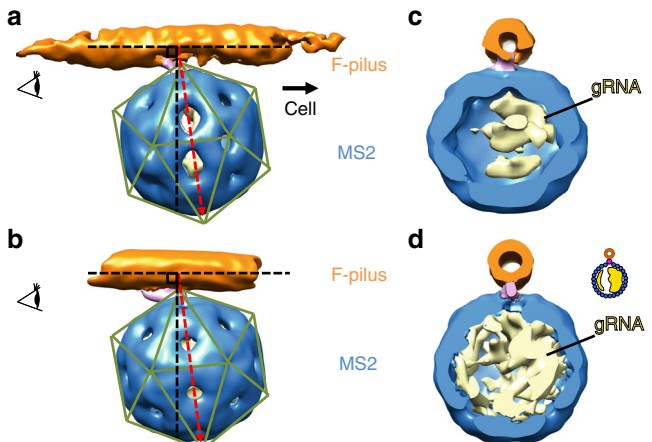

**Fig. 5** Comparison of the subtomogram average and the single-particle cryo-EM map. **a** A side view of the subtomogram average of the MS2/F-pilus complex with the direction to the cell envelope labeled by a solid black arrow. **b** A side view of the low-pass filtered single-particle cryo-EM map of the MS2/F-pilus complex at 30-Å resolution. The axis of the F-pili is labeled by a horizontal dashed black line with another dashed black line perpendicular to it. A dashed red arrow indicates the two-fold axis on the side of the near-icosahedral capsid. Green lines indicate the icosahedral lattice of the capsid. **c**, **d** Cut-in views of the subtomogram average map and the low-pass filtered single-particle cryo-EM map of the MS2/F-pilus complex viewed from the orientation labeled by the eye cartoons in panels **a** and **b**. The inset shows a cartoon model for the asymmetric distribution of the gRNA

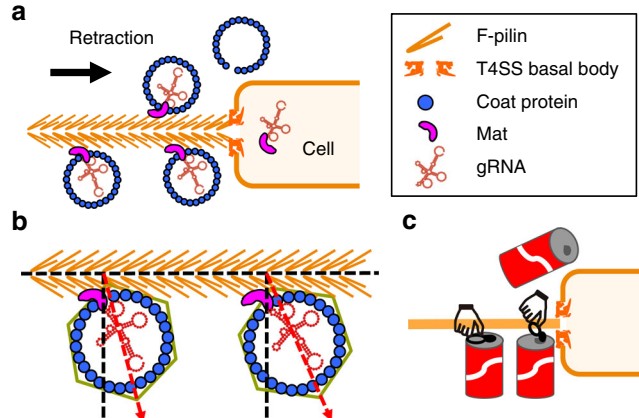

**Fig. 6** A model for adsorption of MS2 to the host. **a** MS2 adsorbed to the side of the F-pilus with the tip of the β-region facing opposite to the pilus retraction. As the F-pilus retracts, MS2 virions are brought closer to the cell envelope. Finally, the Mat-gRNA complex enters the cytoplasm. **b** Binding of the F-pilus causes slight orientational variations of the Mat relative to the rest of the MS2 capsid. **c** The orientation of the Mat may facilitate its detachment from the capsid just as opening a pop tab on a coke can

plasmid was confirmed by colony PCR and sequencing. The purification protocol for F-pili is optimized from the previously published method[23]. Briefly, *E. coli* MC4100 cells were grown on 100 Luria-Bertani (LB) plates with kanamycin 5 μg/ml for the F-pili production for 16 h. The *E. coli* cells were scraped from the surface of the plates with pre-chilled SSC buffer (15 mM sodium citrate, 150 mM NaCl, pH 7.2), then passed through 25 1/2 G needles for three times. The suspension was collected and incubated at 4 °C for 4 h with gentle stirring. The cells were pelleted out by centrifugation. Five percent PEG 6000 were gradually added into the supernatant fraction and the NaCl concentration was adjusted to 500 mM. Then the fraction was incubated at 4 °C for 4 h with gentle stirring. The pili were precipitated by centrifugation at 15,000g for 50 min and then resuspended in 1 ml of the F-pili buffer (50 mM Tris-HCl, 200 mM NaCl, pH 8.0). The suspension was layered on pre-formed CsCl step gradients (1.1–1.3 g/cm³) and centrifuged at

192,000g for 19 h at 4 °C. The band for F-pili was dialyzed against the F-pili buffer. MS2 were amplified by infecting *E. coli* ER2738. Phage particles were precipitated from phage lysate using ammonium sulfate and purified with CsCl gradient[34]. The final concentration is about ~5 mg/ml of the phage proteins. The titer is ~$10^{12}$ plaque-forming units.

**Single-particle cryo-EM specimen preparation and imaging**. Phage MS2 and F-pili were incubated for 30 min at 37 °C for adsorption. Three microliters of the sample was applied to a 400-mesh 1.2/1.3 c-flat copper grid and frozen with a Vitrobot Mark III (FEI Company). The sample was imaged under JEM3200FSC cryo-electron microscope operated at 300 kV. Movie stacks were collected using SerialEM[35] in the manual mode with a Gatan K2 Summit direct detection camera (Gatan) at a nominal magnification of ×30,000 in the super-resolution mode, leading to a sub-pixel size of 0.615 Å. A total electron dose of ~37 $e^-$/Å$^2$ was fractionated over 40 frames (0.2 s/frame, 8 s in total). An in-column energy filter was used with a slit width of 30 eV. To address the preferred-orientation problem, we used the tilting strategy[25]. In total, 4195 super-resolution movie stacks were obtained, with 1,490, 1,241, and 1,464 stacks collected at 0°, 15°, and 30° tilting, respectively.

**Single-particle cryo-EM image processing**. The super-resolution movie stacks were binned by 2 then aligned and filtered by Motion Correction 2 (ref. [36]). In total, 168,989 particles with good quality were semi-automatically picked from 4,195 micrographs by EMAN2 (ref. [37]). Gctf was used to estimate the contrast transfer function for each particle[38]. Then the unsupervised refinement was performed in Relion[29] with the C1 symmetry without any mask, which yielded a map of the MS2/F-pilus complex (with the gRNA). The refinement was also performed with a mask around the F-pilus and the MS2 capsid and yielded the 5.0 Å-resolution map of the MS2/F-pilus complex (without the gRNA). This refinement was continued with a loose mask around the Mat and the F-pilus, which resulted in more uniform resolutions around the region of the Mat and the F-pilus. The multibody refinement was performed with the F-pilus and MS2 in their respective masks, followed by the principal component analysis to analyze the most dominant structural variations within the structure[28]. To further explore the flexibility in the MS2/F-pilus complex, we also continued the refinement from the MS2/F-pilus complex (with gRNA) reconstruction with a spherical mask of 280 Å in diameter to further align the phage particles. We then did the Relion 3D classification with a loose mask of the F-pilus without alignment. The particles were classified into four classes with 34.38% in Class 1, 33.29% in Class 2, 11.99% in Class 3, and 20.34% in Class 4. Refinements were then performed separately for Classes 1–4 with a mask around the F-pilus and the MS2 capsid. Class 4 was a junk class showing the F-pilus as two layers due to severe preferred orientations.

**Single-particle cryo-EM imaging and processing of apo MS2**. The apo-state MS2 was imaged using an FEI TF20 cryo-electron microscope operated at 200 kV equipped with a Gatan K2 Summit direct detection camera at a nominal magnification of ×29,000 in the super-resolution mode, yielding a sub-pixel size of 0.625 Å. In all, 1,329 movie stacks were collected using SeilEM in the automated mode. For each movie stack, a total dose of ~34 $e^-$/Å$^2$ was fractionated over 33 frames with 0.2 s per frame and 6.6 s in total. A total of 44,987 particles were picked and refined by Relion, produced a density map at 6.6 Å resolution. To investigate the flexibility in the Mat, the same 3D Classification strategy, used in the pilus-bound state, was performed with a loose mask around the Mat region with the alignment skipped. Almost all (93%) of the particles went into the same class, indicating a relatively homogeneous conformation in this region of the phage particle.

**Molecular modeling and map visualization**. To build the model for the Mat/F-pilus complex, the PDB models (accession number 5LER and the Chain M of 5TC1) were first fit into the density map and used as the initial models for the F-pilus and the Mat, respectively. The Cα of the tip region for the Mat was manually traced in the density using Coot[39]. Then the model of the complex was refined by the real-space refinement in Phenix[40]. The statistics of the refined model is described in Supplementary Table 1. To compare the internal plasticity within the Mat from three different conformations, the same initial model of the Mat was refined against the density maps for Classes 1, 2, and 3, respectively, using MDFF[41]. The map segmentation and visualization were done in UCSF Chimera[42]. It has been reported that the wild-type F-pili have two helical-rise parameters of 13.2 and 12.5 Å[23]. Such a difference only leads to less than a 2-Å shift, between the two types of helical symmetry, at the distal ends within the binding site of a single Mat and may not affect the interaction points which occur in the center of the binding interface.

**Specimen preparation for cryo-ET**. *E. coli* F$^+$ strain MC4100 with the pOX38 (Kanamycin) plasmid was grown in LB liquid culture. Then the cell pellet was spun down and resuspended in the MS2 buffer, which was pre-warmed to 37 °C. MS2 phages were incubated with the cells for 30 min at 37 °C for adsorption. The 6nm-diameter gold particles were added as fiducial markers for image alignment. Three microliters of the sample was applied on the Quantifoil 3.5/1 grids and frozen with the Vitrobot Mark III (FEI Company).

**Cryo-ET data collection and subtomogram averaging**. Frozen-hydrated specimens were imaged at −170 °C, using a Polara G2 electron microscope (FEI) equipped with a field emission gun and a Gatan K2 Summit direct detection camera. The microscope was operated at 300 kV with a magnification of ×9,400 with an effective pixel size of 4.45 Å without binning. Using SerialEM, low-dose, single-axis tilt series were collected at −6 to −9 μm defocus with a cumulative dose of ~80 $e^-$/Å$^2$ distributed over 35 stacks and covering an angular range of −51° to +51°, with an angular increment of 3°. Each stack contains eight frames. We used Tomoauto[43] to facilitate data processing which includes drift correction of dose-fractionated data using Motioncorr[44], assembly of corrected sums into tilt series, automatic fiducial seed model generation, alignment and contrast transfer function correction of tilt series by IMOD[45], and reconstruction of tilt series into tomograms by Tomo3D[46]. In total, 100 tomographic reconstructions were generated. To make sure the pili are directly connected with *E. coli*, a total of 1,245 subtomograms (400 × 400 × 400 voxels) were visually identified and then extracted from 30 tomographic reconstructions in which the pili are extended from the *E. coli* cell membrane. Then we used the tomographic package I3 (0.9.9) for subtomogram analysis as described previously[47]. The initial orientation of each particle was estimated by the phage and pili coordinates, thereby providing the three Euler angles. To accelerate image analysis, 4 × 4 × 4 binned subtomograms (100 × 100 × 100 voxels) were used for initial alignment. The alignment proceeded iteratively with each iteration consisting of three parts in which references and classification masks were generated, subtomograms were aligned and classified, and finally class averages were aligned with each other. After multiple cycles of alignment and classification for the 4 × 4 × 4 binned subtomograms, we used the 2 × 2 × 2 binned subtomograms for refinement. Fourier shell correlation (FSC) between the two independent reconstructions was used to estimate the resolution of the averaged structures.

**Reporting Summary**. Further information on research design is available in the Nature Research Reporting Summary linked to this article.

## Data Availability

The cryo-EM and cryo-ET density maps of the MS2/F-pilus complex and the apo MS2 have been deposited into the EMData Bank with accession numbers EMD-9397, EMD-0453, EMD-9399, EMD-0448, EMD-0450, EMD-0338, and EMD-0451. The model of the F-pilus/maturation protein complex has been deposited to the protein data bank with the accession ID 6NM5 [https://doi.org/10.2210/pdb6NM5/pdb]. The authors declare that all data supporting the findings of this study are available from the corresponding authors upon request.

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

## Acknowledgements

We thank the Microscopy and Imaging Center at Texas A&M University for providing instrumentation for the initial screening of the phage particles and the data collection for the apo-state MS2; the Texas A&M High-Performance Research Computing Center for providing the computational resources for the data processing; and the National Center for Macromolecular Imaging (NCMI) at Baylor College of Medicine for single-particle cryo-EM data collection of the MS2/F-pilus complex. NCMI is supported by NIH Grants P41GM103832 and U24GM1167. J.Z. is supported by start-up funding from the Department of Biochemistry and Biophysics at Texas A&M University. J.Z. acknowledges the support of the Center for Phage Technology, jointly sponsored by Texas AgriLife and Texas A&M University, and the NIH grants R21AI137696, P01AI095208. J.Z. and B.H. acknowledge the Welch Foundation grants A-1863 (to J.Z.) and AU-1953 (to B.H.). R.M. and M.J. are partly supported by the Texas A&M Triads for Transformation grant, the TAMU School of Science Strategic Transformative Research Program and the CST*R Foundation. *E. coli* F+ strain MC4100 containing the pOX38 plasmid was a gift from Dr. Peter J. Christie. We also thank Dr. Peter J. Christie, Dr. Ry Young, and Karl V. Gorzelnik for insightful critiques on the manuscript.

## Author Contributions

R.M., B.H. and J.Z. designed research; R.M., M.J., Z.C., J-Y.C., K.Y., B.H. and J.Z. performed research with the assistance of J.J., X.Y., and Z.W.; R.M., B.H., and J.Z. analyzed data; and R.M., B.H. and J.Z. wrote the paper.

## Additional information

**Competing interests:** The authors declare no competing interests.

