## [Peer Review File · Nature Communications]

Reviewers' comments:

Reviewer #1 (Remarks to the Author):

Single-stranded RNA bacteriophages infect gram-negative bacteria by absorbing to the side of retractile pilus of the host cell via the Mat protein. Mat and genome RNA (gRNA) are then taken up together by the host cell. The structural and molecular mechanisms for the pilus absorption and host infection are still not well known. In this manuscript, Meng et al investigated the molecular interaction between MS2 phage and F-pilus by single-particle cryo-EM, and determined the polarity of pilus and phage Mat protein relative to the host envelope using cryo-ET and subtomogram averaging method. Combining the information from this manuscript and previous knowledge, the authors provided a possible “pop tab” model for Mat-gRNA complex entering into the host cell. The manuscript is well written and conclusions are in general supported by experimental data. Although the resolution of reported structures is still limited, the manuscript provides many important pieces of information for the infection mechanism.

Overall, I recommend the publication of the manuscript.

There are some major and minor concerns about the manuscripts.

Major concerns:

1. A major conclusion is the electrostatic and hydrophobic interactions at the pilus-Mat interface, which is based on the side-chain interaction analysis. However, the resolutions of the three conformational states are ranging from 5.6 Å to 7.3 Å, and the local refinement of the pilus-Mat complex did not result in significant improvement (at a resolution of 6.2 Å). Although the interface analysis seems to be supported by previous mutational data and sequence conservation among difference species. Given the resolutions reported, the authors should indicate that these interactions are based on main chain positions (or lowering the tone in these statement). Also, the first five residues AGSSG of pilin are not that hydrophobic. The authors should clarify this in the manuscript.
2. For the same reason in 1, in the first three lines of Page 6, the authors stated that RMSDs between different Mat conformers are very small. Since the models were derived by MDFF, the numbers reported here may have been beyond the precision / accuracy of MDFF method.
3. Another major conclusion in this manuscript is the binding of the F-pilus destabilizes the maturation protein from the capsid, which is based on the single particle cryo-EM results that a

fraction (Class3, ~12%) of particles show relative larger angle between Mat and phage capsid. It implies that Mat proteins could change its orientation relative to phage capsid. It does not necessarily mean that their interactions might have been weakened. Given the infection model, premature detachment of gRNA-Mat complex from virions might not be good for infection?

Minor concerns:

1. In the second paragraph, page 4, the resolutions for different conformations should range from 5.6 to 7.3 Å. The resolution 5 Å is for all the particles, not for a conformational state.
2. In Figure S4, page 25, the color codes for panel A and B are not consistent, but only stated in panel A. Besides, the description "Phospholipids are colored by element" is confusing and the elements of phospholipids couldn't be distinguished.
3. As stated in the second paragraph, page 5, "The gRNA inside the capsid of the F-pilus-bound MS2 is in the same conformation as previously reported for the free virions". However, there are no labels for gRNA in any figures. The labels for gRNA should be added in related figures.
4. Both "β-region of mat" and "tip of mat" occurred throughout the manuscript, and they seemed to refer to the same region of Mat. The authors should define the two terms and label them in related figures.
5. In the second paragraph, page 10, the resolution of apoMS2 is stated as 6.6 Å. But in Figure S3, it was labelled as 6.7 Å. Also, the authors used more than 40K particles for apo-MS2 structural determination. With these particles, one would expect a map far better than 6.6 Å. Are there any limiting factors in Apo structure determination?

Reviewer #2 (Remarks to the Author):

REVIEW

In this manuscript by Meng et al., the authors have performed cryo-electron microscopy (cryoEM) in both single particle analysis mode and cryo-electron tomography (cryoET) mode to visualize the adsorption of ssRNA viral particles onto the pili of *E. coli* in vitro and in situ. This adsorption is the primary mode of this virus's infection into the bacteria; the viral ssRNA gets uptaken by the bacteria when the pilus, onto which the viral particle is adsorbed, is retracted into the bacterial cell. This

adsorption is mediated by a viral protein called Mat which sits on the tip of the viral particles by replacing one of the dimers of the viral capsid protein. The Mat is tightly bound to the ssRNA packaged inside the viral particle. It has been biochemically shown that Mat forms tight interaction with the pilus of the bacteria, thus allowing the entire viral particle to be tightly adsorbed onto the pilus. When the pilus is retracted, the viral particle is also retracted and likely hits the envelope of the bacteria to disintegrate. The ssRNA exposed due to this disintegration, gets uptaken by the bacterial cell through an unknown pathway.

The amount of experiments, data analysis and data generated is quite impressive. This manuscript represents a significant shift from the current literature on the structural analysis of phage infection via the retractable pili of bacterial microbes. The experiments are expertly performed and the manuscript is well written; I highly recommend the publication of this manuscript. I have a few questions which I think will enhance the manuscript further. I leave it to the authors and editors on how many of these questions or their answers they would like to incorporate into the manuscript. Although, some of these questions need to be addressed before the publication (e.g., #1, #2, #5, maybe #8 in methods section?),.

1. In Fig.1C, it is quite hard to make out the arrangement of a1, a2 and a3 segments of the pilin in the EM density. I can't easily pin-point a1, a2 or a3 segments in the EM density. The authors should draw this schematic better

2. One of the primary messages of the manuscript is that the Mat protein is destabilized when the viral particle adsorbs onto the pilus. But this message is not explicitly stated and explained in the manuscript. The authors should make this more obvious. The only good reference to this message is the following statement(s) :

'Such a variation for the positioning of the Mat is not observed in our control experiment, imaging the pilus-free MS2'

3. There is always an active play between the bacterial strain and the phages leading to an active equilibrium between them. Due to the active nature of this equilibrium, do you expect the sequences of the amino acids that are involved in the interaction of the Mat with the pilus to be hypervariable? Is it hypervariable or quite conserved?

4. How tightly is ssRNA packed into the viral particle? And how many ssRNA molecules are there inside each viral particle? one or many? And what is the average length of the ssRNA?

5. Using the asymmetric distribution of ssRNA inside the viral particle has been vital to both the data analysis and is also one of the main messages of the manuscript, it would be good to have a panel dedicated to showing the structures that show the asymmetric distribution and its relation with the interface of interaction between Mat and pilus. This feature can be ascertained from

current panels, but it would be good to have a dedicated panel for this in any of the main text figure or SI figure.

6. It would be good to see some speculation about the reason(s) behind the asymmetric distribution of ssRNA inside the viral particle?

7. There is a significant relative motion between the pilus and the viral particle(s). Which one moves the most? The pilus or the viral particles? Any speculations about this?

8. Could you have gone higher than 80 e-/Å² dose for cryoET?

9. Which interaction interface between Mat and pili is the most crucial for their tight binding? It would be crucial for other scientists to target such interaction interfaces to develop anti-viral resistance for these strain of bacterial microbes against viruses like this ssRNA one.

10. What would be the most effective strategy to capture these viral particles as they are being retracted onto the bacterial cell along with the pili, in the cryoET pipeline?

Reviewers' comments:

Reviewer #1 (Remarks to the Author):

Single-stranded RNA bacteriophages infect gram-negative bacteria by absorbing to the side of retractile pilus of the host cell via the Mat protein. Mat and genome RNA (gRNA) are then taken up together by the host cell. The structural and molecular mechanisms for the pilus absorption and host infection are still not well known. In this manuscript, Meng et al investigated the molecular interaction between MS2 phage and F-pilus by single-particle cryo-EM, and determined the polarity of pilus and phage Mat protein relative to the host envelope using cryo-ET and subtomogram averaging method. Combining the information from this manuscript and previous knowledge, the authors provided a possible "pop tab" model for Mat-gRNA complex entering into the host cell. The manuscript is well written and conclusions are in general supported by experimental data. Although the resolution of reported structures is still limited, the manuscript provides many important pieces of information for the infection mechanism.

Overall, I recommend the publication of the manuscript.

There are some major and minor concerns about the manuscripts.

Response: We thank the reviewer for the insightful critiques and suggestions, which we will address as follows.

Major concerns:

1. A major conclusion is the electrostatic and hydrophobic interactions at the pilus-Mat interface, which is based on the side-chain interaction analysis. However, the resolutions of the three conformational states are ranging from 5.6 Å to 7.3 Å, and the local refinement of the pilus-Mat complex did not result in significant improvement (at a resolution of 6.2 Å). Although the interface analysis seems to be supported by previous mutational data and sequence conservation among difference species. Given the resolutions reported, the authors should indicate that these interactions are based on main chain positions (or lowering the tone in these statement). Also, the first five residues AGSSG of pilin are not that hydrophobic. The authors should clarify this in the manuscript.

Response: We thank the reviewer for the suggestions and have lowered the tone in the manuscript as follows:

(1) We have revised Panel B-D of Figure 3 and the figure legends to highlight the main-chains of the interacting residues instead of the side-chains by using bead models to represent the C-alpha atoms. Please see the figure below.

Figure 3. The molecular interface between MS2 and the F-pilus. (A) Four F-pilin subunits, which have specific interactions with the Mat, are colored tan, orange, gold, and khaki, respectively. The inset shows the overall side view of the F-pilus/Mat complex and the location of these four F-pilins within the pilus. Three aspartic acids are labeled as sphere models and the N-terminus of one pilin is labeled by a black sphere. Arrows indicate the viewing directions for Panels B-D. (B) The negatively charged residue Asp7 (red bead) from one F-pilin is in close proximity to the positively charged His357 (blue bead) from the Mat. (C) The negatively charged residues Asp7 and Asp23 (red beads) from two F-pilins are in close proximity to Arg36 and Arg99 (blue beads) from the Mat, respectively. (D) The N-terminus of one pilin extends into a hydrophobic pocket consisting of five hydrophobic residues (Val15, Phe31, Leu33, Phe92, and Phe94, labeled by grey beads) from the Mat. The dashed black line denotes the potential location of the five missing residues (AGSSG) at the N-terminus of one pilin. The cryo-EM density map is shown as a grey transparent envelope. "

(2) We have revised the main text of the manuscript, which are highlighted in dark red.

We thank the reviewer for pointing out that the five residues AGSSG of pilin are not that hydrophobic, with only the first residue, Ala, contributing to the hydrophobicity. However, further calculation of the surface hydrophobicity reveals a strong hydrophobic pocket on the Mat, into which the Ala of an AGSSG loop extends. Such a pocket consists of the previously labeled Phe92 and Phe94, as well as three more hydrophobic residues, Val15, Phe31 and Leu33, which are conserved in MS2-like phages. The Mat of Phage Fr lacks Phe92 and Phe94, but still has Val15, Leu31, and Leu33, partially preserving the hydrophobic pocket.

Therefore, even though the AGSSG loop is not that hydrophobic, the strong hydrophobic pocket from the Mat may still interact with it. This interaction is also supported by the fact that the pED208 variant of the F-pilus interacts weakly with MS2, presumably due to the lack of the AGSSG loop.

We have made a new Panel C of Fig. S4 to show this hydrophobic pocket on the Mat, as well as a revised Fig. S1B to color label the residues Val15, Phe31/Leu31, and Leu33 (see below). We have also revised the main text of the manuscript, which are highlighted in dark red.

Figure S4. The structure of the F-pilus, bound with MS2, is consistent with the previously published structure of the apo F-pilus. (A) Phospholipids are visible in the map of the F-pilus/Mat complex. Pilin proteins and the Mat are colored in white. Phospholipids are colored by element. Oxygen, phosphorus, and carbon are shown in red, orange, and gray, respectively. **(B)** The Ala18 residues exposed at the pilus-Mat interface are labeled by gray spheres. The α -region, the β -region, and the tip of the β -region are labeled by brackets and a black circle, respectively. **(C)** N-terminus of one pilin (labeled by a black sphere at the end of a yellow ribbon model) extends into a hydrophobic pocket (enclosed by a black oval) consisting of Val15, Phe31, Leu33, Phe92, and Phe94 from the Mat. The dashed black line denotes the potential location of the five missing residues (AGSSG) at the N-terminus of one pilin. The surface of the Mat are colored from dodger blue for the most hydrophilic, to orange red for the most hydrophobic. "

“Figure S1. The sequence alignment for the F-like pilins and the Mat of MS2-like phages. (A) The sequence comparison among five F-like pilins. (B) The sequence comparison among the Mats of MS2 and MS2-like phages. The residues involved in the pilus/Mat interactions or indicated in previous mutagenesis studies are labeled blue (positively charged), red (negatively charged) or orange (AGSSG loop or hydrophobic). Residue 31 in the Mat of Phage Fr is a Leu instead of a Phe, but still hydrophobic, and is labeled yellow.”

2. For the same reason in 1, in the first three lines of Page 6, the authors stated that RMSDs between different Mat conformers are very small. Since the models were derived by MDFF, the numbers reported here may have been beyond the precision / accuracy of MDFF method.

Response: We thank the reviewer for pointing this out. The point we want to make is that the Mats in the three conformations of the pilus-bound MS2 do not show detectable variations in the tertiary structures. To estimate the fitting precision of MDFF, we calculated the average and standard deviation (error) of the fitted models. We have made a new Figure S7 and revised the text accordingly (see below).

“Figure S7. Structural comparisons of the Mats from Classes 1, 2 and 3 reveal no detectable variation in the tertiary structure. To estimate the fitting precision of MDFP, we first calculated the average and standard deviation (error) for the fitted models as follows. For each of the three conformations, we extracted 200 frames from the MDFP trajectory after the simulation converged. Each frame represents a fitted conformation, which fluctuates around an average conformation. We then estimated the mean and standard deviation (error) for the position of each C α atoms among these 200 models for each conformation. The standard deviations of the C α position are colored-coded on the model in Panel A and between 0.25 Å (blue) ~1.24 Å (red), which are smaller than the resolution of the map (5.6-7.3 Å). This is due to the fact that the MDFP fitting uses not only the information from the density map but also the molecular energy, which maintains a stereochemically correct and stable structure. We then estimated the difference among the three conformations by calculating the C α deviation between each pair of the average conformations. Panel B shows the C α deviations between each pair of conformations, color-coded on the model, which are between 0.06 Å (blue) ~ 4.63 Å (red), with the largest deviations in the flexible loop regions. Overall, the models of the Mat in the three conformations are very similar with most of the regions showing a C α deviation of less than 1 Å. Moreover, the direct overlapping of the density maps of the three conformations, in Panel C, shows high cross correlations of 0.97 between Class1 and Class2, 0.96 between Class1 and Class3, and 0.96 between Class2 and Class3, which means the three density maps of the Mat are very similar to each other. In summary, at our current resolutions of the maps, there is no detectable variation in the tertiary structure of the Mat in these three conformations of the pilus-bound MS2.”

3. Another major conclusion in this manuscript is the binding of the F-pilus destabilizes the maturation protein from the capsid, which is based on the single particle cryo-EM results that a fraction (Class3, ~12%) of particles show relative larger angle between Mat and phage capsid. It implies that Mat proteins could change its orientation relative to phage capsid. It does not necessarily mean that their interactions might have been weakened. Given the infection model, premature detachment of gRNA-Mat complex from virions might not be good for infection?

Response: We thank the reviewer to point out the potential confusion in our manuscript. It's true that the premature detachment of the gRNA-Mat is not good for the infection and we are sorry to cause that confusion while what we really meant was the flexibility induced by the pilus binding might prime the gRNA-Mat complex to leave the rest of the capsid. Indeed, in our control experiment with the apo-MS2, such a variation in the orientations for the Mat relative to the neighboring coat proteins was not observed. We have clarified this in our discussion section of our manuscript to avoid the impression that the pilus binding causes the premature detachment of the gRNA-Mat.

Minor concerns:

1. In the second paragraph, page 4, the resolutions for different conformations should range from 5.6 to 7.3 Å. The resolution 5 Å is for all the particles, not for a conformational state.

Response: We have revised the text as follows.

"In this study, we present the single-particle cryo-EM structures of different conformations of the MS2/F-pilus complex at resolutions ranging from 5.6 to 7.3 Å and an average reconstruction with all particles at 5.0 Å resolution. "

2. In Figure S4, page 25, the color codes for panel A and B are not consistent, but only stated in panel A. Besides, the description "Phospholipids are colored by element" is confusing and the elements of phospholipids couldn't be distinguished.

Response: In Fig. S4, we have removed the old Panel B, which is not necessary in the context of the manuscript. We have also revised Panel A to make the phospholipids more apparent (see Page 2 of this letter for the revised figure).

3. As stated in the second paragraph, page 5, "The gRNA inside the capsid of the F-pilus-bound MS2 is

in the same conformation as previously reported for the free virions". However, there are no labels for gRNA in any figures. The labels for gRNA should be added in related figures.

Response: We added the "gRNA" label in all the related figures.

4. Both "β-region of mat" and "tip of mat" occurred throughout the manuscript, and they seemed to refer to the same region of Mat. The authors should define the two terms and label them in related figures.

Response: The β-region of mat refers to the β-sheet-rich region of the Mat. The tip of the mat refers to the very tip of the β-region. To avoid this confusion, we have revised the Panel B of Figure S4 to define and label the β-region and the tip of the β-region (see Page 2 of this letter for the revised figure). We have also changed all the "tip of mat" to "tip of the β-region" in the text.

5. In the second paragraph, page 10, the resolution of apoMS2 is stated as 6.6 Å. But in Figure S3, it was labelled as 6.7 Å. Also, the authors used more than 40K particles for apo-MS2 structural determination. With these particles, one would expect a map far better than 6.6 Å. Are there any limiting factors in Apo structure determination?

Response: We thank the reviewer to spot this inconsistency. We have changed the resolution in the text to 6.7Å.

For the control experiment, the Apo MS2 was imaged using an FEI TF20 cryo-electron microscope operated at 200kV. We planned this experiment as a control experiment and the resolution was limited by the experimental setup.

Reviewer #2 (Remarks to the Author):

REVIEW

In this manuscript by Meng et al., the authors have performed cryo-electron microscopy (cryoEM) in both single particle analysis mode and cryo-electron tomography (cryoET) mode to visualize the adsorption of ssRNA viral particles onto the pili of E. coli in vitro and in situ. This adsorption is the primary mode of this virus's infection into the bacteria; the viral ssRNA gets uptaken by the bacteria when the pilus, onto which the viral particle is adsorbed, is retracted into the bacterial cell. This adsorption is mediated by a viral protein called Mat which sits on the tip of the viral particles by replacing one of the dimers of the viral capsid protein. The Mat is tightly bound to the ssRNA packaged inside the viral particle. It has been biochemically shown that Mat forms tight interaction with the pilus of the bacteria, thus allowing the entire viral particle to be tightly adsorbed onto the pilus. When the pilus is retracted, the viral particle is also retracted and likely hits the envelope of the bacteria to disintegrate. The ssRNA exposed due to this disintegration, gets uptaken by the bacterial cell through an unknown pathway.

The amount of experiments, data analysis and data generated is quite impressive. This manuscript represents a significant shift from the current literature on the structural analysis of phage infection via the retractable pili of bacterial microbes. The experiments are expertly performed and the manuscript is well written; I highly recommend the publication of this manuscript. I have a few questions which I think will enhance the manuscript further. I leave it to the authors and editors on how many of these questions or their answers they would like to incorporate into the manuscript. Although, some of these questions need to be addressed before the publication (e.g., #1, #2, #5, maybe #8 in methods section?).

Response: We thank the reviewer for the support of our work and will address the questions below.

1. In Fig.1C, it is quite hard to make out the arrangement of a1, a2 and a3 segments of the pilin in the EM density. I can't easily pin-point a1, a2 or a3 segments in the EM density. The authors should draw this schematic better.

Response: We have revised the Panel C of Figure 1 by coloring and labeling a single copy of the F-pilin as follows.

Figure 1. The overall architecture of the MS2/F-pilus complex. (A) An electron micrograph of the MS2/F-pilus complex. The scale bar denotes 500 Å. (B) The side view of the MS2/F-pilus complex with the F-pilus colored in orange, the Mat colored in pink, coat proteins in blue and gRNA in light yellow. For visualization, the density map in Figure 1 is a composite map with the density of the F-pilus/Mat complex replaced by the 6.2 Å-resolution map, which has a more uniformly well-resolved F-pilus fragment. The axis of the F-pilus is labeled by a horizontal dashed black line with another vertical dashed black line perpendicular to it. A dashed red arrow denotes the two-fold axis on the side of the MS2 capsid. The angle between the vertical dashed black line and the dashed red arrow is 7.8°. (C) The model of the Mat/F-pilus complex fit in the cryo-EM map. One of the F-pilin is colored from purple (N terminus) to green (C terminus) as in the inset to show the orientation of the F-pilins in the assembled pilus. The three α -helices of an F-pilin are also labeled in the inset. (D) Cut-in view of the density map of the MS2/F-pilus complex viewed from the direction of the eye cartoon in Panel B. The outer and inner diameters of the F-pilus are 87 Å and 28 Å, respectively. The diameter of the MS2 capsid is ~250 Å. The density of the gRNA is from the refinement of all particles without any mask and low-pass filtered to 8 Å. The inset shows a cartoon model for the asymmetric distribution of the gRNA.

2. One of the primary messages of the manuscript is that the Mat protein is destabilized when the viral particle adsorbs onto the pilus. But this message is not explicitly stated and explained in the manuscript. The authors should make this more obvious. The only good reference to this message is the following statement(s): 'Such a variation for the positioning of the Mat is not observed in our control experiment, imaging the pilus-free MS2'

Response: We have revised our discussion section of the manuscript to explicitly state our observation.

"Finally, we observed that the binding of the F-pilus causes the variations in the orientations for the Mat relative to the neighboring coat proteins (**Fig. 6B**). As the Mat is still attached to the gRNA, the flexibility induced by the pilus binding may prime the Mat-gRNA complex to leave the rest of the capsid."

3. There is always an active play between the bacterial strain and the phages leading to an active equilibrium between them. Due to the active nature of this equilibrium, do you expect the sequences of the amino acids that are involved in the interaction of the Mat with the pilus to be hypervariable? Is it hypervariable or quite conserved?

Response: According to the sequence comparison in Fig S1, the protein sequences of the pilin are quite conserved. The Maturation protein sequences have some variations, but the critical amino acids involved in the interactions are conserved.

4. How tightly is ssRNA packed into the viral particle? And how many ssRNA molecules are there inside each viral particle? one or many? And what is the average length of the ssRNA?

Response: Unlike dsDNA phages, which have a tightly packed genomic DNA, ssRNA are sparsely packed inside the capsid. There is only one ssRNA molecule inside each viral particle with a length of 3569bp. We have added this information in the introduction section of our manuscript

5. Using the asymmetric distribution of ssRNA inside the viral particle has been vital to both the data analysis and is also one of the main messages of the manuscript, it would be good to have a panel dedicated to showing the structures that show the asymmetric distribution and its relation with the interface of interaction between Mat and pilus. This feature can be ascertained from current panels, but it would be good to have a dedicated panel for this in any of the main text figure or SI figure.

Response: We have added a cartoon inset in Figures 1, Figure 5 and Figure S5 to indicate the asymmetry of the RNA distribution inside the capsid. For Figure 1, please see Page 6 of this letter. For Figures 5 and S5, please see below.

Figure 5. Comparison of the subtomogram average and the low-pass filtered single-particle cryo-EM map. (A) A side view of the subtomogram average of the MS2/F-pilus complex with the direction to the cell envelope labeled by a solid black arrow. (B) A side view of the low-pass filtered single-particle cryo-EM map of the MS2/F-pilus complex at 30-Å resolution. The axis of the F-pilus is labeled by a horizontal dashed black line with another dashed black line perpendicular to it. A dashed red arrow indicates the two-fold axis on the side of the near-icosahedral capsid. Green lines indicate the icosahedral lattice of the capsid. (C, D) Cut-in views of the subtomogram average map and the low-pass filtered single-particle cryo-EM map of the MS2/F-pilus complex viewed from the orientation labeled by the eye cartoons in Panels A and B. The inset shows a cartoon model for the asymmetric distribution of the gRNA.

“Figure S5. The low-pass filtered single-particle cryo-EM map showing asymmetric features. (A, B) Side views of the low-pass filtered single-particle cryo-EM maps of the MS2/F-pilus complex at 60-Å (panel A) and 80-Å (panel B) resolutions showing a tilting angle between the pilus axis and the two-fold symmetry axis of the capsid. The axis of the F-pilus is labeled by a horizontal dashed black line with another dashed black line perpendicular to it. The dashed red arrow indicates the two-fold axis on the side of the near-icosahedral capsid. Green lines indicate the icosahedral lattice of the capsid. **(C, D)** Cut-in views of the low-pass filtered single-particle cryo-EM map of the MS2/F-pilus complex viewed from the orientation labeled by the eye cartoons in Panels A and B, showing the unevenly distributed gRNA density. The inset shows a cartoon model for the asymmetric distribution of the gRNA.”

6. It would be good to see some speculation about the reason(s) behind the asymmetric distribution of ssRNA inside the viral particle?

Response: The RNA is sparsely packed inside the capsid and does not follow the icosahedral symmetry of the capsid. Such an asymmetry in the gRNA packing is formed as the capsid proteins assemble around the gRNA and the gRNA folds. We have added this explanation of the asymmetry of RNA distribution in the main text.

7. There is a significant relative motion between the pilus and the viral particle(s). Which one moves the most? The pilus or the viral particles? Any speculations about this?

Response: In our single-particle cryo-EM experiment, the pilus-MS2 complex is reconstituted *in vitro* and the movement between them is relative. However, *in vivo*, we expect the F-pili are more dynamic as it has to retract and spin while MS2 particles tightly bind to them (Clarke, M., et al., *F-pili dynamics by live-cell imaging*. Proc Natl Acad Sci U S A, 2008. **105**(46): p. 17978-81.).

8. Could you have gone higher than 80 e-/Å² dose for cryoET?

Response: Frozen-hydrated biological samples are very sensitive to the electron beam and deteriorate gradually upon increasing exposure to the beam. Generally, the total dose for an entire tilt series should be no more than 100 e-/Å², otherwise visible bubbling will occur on the recorded area. So, typically a total dose of 60-100 e-/Å² can be applied to bacterial samples. However, this is a rough guideline and some samples require a lot less electron exposure, especially when attempting to reconstruct a high-resolution map.

9. Which interaction interface between Mat and pili is the most crucial for their tight binding? It would be crucial for other scientists to target such interaction interfaces to develop anti-viral resistance for these strain of bacterial microbes against viruses like this ssRNA one.

Response: Based on our structural results, the interactions between the Mat and the F-pilus involve a combination of different types of interactions such as the electrostatic interaction, the hydrophobic interaction, etc. These critical residues are shown in our Figure 3 and we would suggest these residues as targets for the mutagenesis studies.

10. What would be the most effective strategy to capture these viral particles as they are being retracted onto the bacterial cell along with the pili, in the cryoET pipeline?

Response: There are only 1-2 F-pili per *E. coli* cell and not each *E. coli* cell shows an associated F-pilus. To improve the efficiency of data collection, first, we acquired montages ("grid square map") at a low magnification (e.g. 2300×). This map provided visible details of pili and their relative positions to the *E. coli* cell. Second, the *E. coli* cells associated with the pili were selected and tilt series were collected at a high magnification (e.g. 9400×). Fortunately, the MS2 particles bind to the F-pilus quite tightly. Therefore, each pilus always showed many phage particles on it.

REVIEWERS' COMMENTS:

Reviewer #1 (Remarks to the Author):

All my comments were well addressed.

Another suggestion for Fig S4a. The colors for phospholipids are still difficult to distinguish. Gray is too similar to the background color, and the orange is too close to red.

Reviewer #2 (Remarks to the Author):

The authors have addressed my questions and concerns. In my opinion, this manuscript is suitable for publication now.

REVIEWERS' COMMENTS:

Reviewer #1 (Remarks to the Author):

All my comments were well addressed.

Another suggestion for Fig S4a. The colors for phospholipids are still difficult to distinguish. Gray is too similar to the background color, and the orange is too close to red.

Response: We thank the reviewer for the suggestion. We have revised Supplementary figure 4 Panel a to make the phospholipids more apparent. Please see the figure below. The revised supplementary figure legend is highlighted in dark red.

“Supplementary figure 4. The structure of the F-pilus, bound with MS2, is consistent with the previously published structure of the apo F-pilus. (a) Phospholipids are visible in the map of the F-pilus/Mat complex. Pilin proteins and the Mat are colored in white. Phospholipids are colored by

element. Oxygen, phosphorus, and carbon are shown in red, gold, and blue, respectively. (b) The Ala18 residues exposed at the pilus-Mat interface are labeled by gray spheres. The α -region, the β -region, and the tip of the β -region are labeled by brackets and a black circle, respectively. (c) N-terminus of one pilin (labeled by a black sphere at the end of a yellow ribbon model) extends into a hydrophobic pocket (enclosed by a black oval) consisting of Val15, Phe31, Leu33, Phe92, and Phe94 from the Mat. The dashed black line denotes the potential location of the five missing residues (AGSSG) at the N-terminus of one pilin. The surface of the Mat is colored from dodger blue for the most hydrophilic, to orange red for the most hydrophobic.”

Reviewer #2 (Remarks to the Author):

The authors have addressed my questions and concerns. In my opinion, this manuscript is suitable for publication now.